# New Radiological Corticalization Index as an Indicator of Implant Success Rate Depending on Prosthetic Restoration—5 Years of Follow-Up

**DOI:** 10.3390/diagnostics14090867

**Published:** 2024-04-23

**Authors:** Tomasz Wach, Jakub Okulski, Rafał Zieliński, Grzegorz Trybek, Adam Michcik, Marcin Kozakiewicz

**Affiliations:** 1Department of Maxillofacial Surgery, Medical University of Lodz, 113 Żeromskiego Str., 90-549 Lodz, Poland; jakub.okulski@gmail.com (J.O.); bkost@op.pl (R.Z.); marcin.kozakiewicz@umed.lodz.pl (M.K.); 2Department of Oral Surgery, Pomeranian Medical University in Szczecin, 70-111 Szczecin, Poland; g.trybek@gmail.com; 34th Military Clinical Hospital in Wroclaw, ul. Rudolfa Weigla 5, 50-981 Wroclaw, Poland; 4Department of Maxillofacial Surgery, Medical University of Gdansk, Mariana Smoluchowskiego 17, 80-214 Gdansk, Poland; adammichcik@gumed.edu.pl

**Keywords:** prosthetic restoration, dental implant, long-term results, long-term success, marginal bone loss, functional loading, intra-oral radiographs, radiomics, texture analysis, corticalization, bone remodeling

## Abstract

The new Radiological Corticalization Index (CI) is an indicator that describes bone remodeling near the dental implant’s neck at the pixel level and is not visible to the naked eye. The aim of this research was to evaluate the correlation between the CI and bone remodeling using only radiographic (RTG) images. RTG samples were divided into groups depending on prosthetic restoration; the implant neck area around dental implants was examined, and texture features of the RTG images were analyzed. The study also investigated the type of prosthetic restoration and its influence as a factor on bone structure. The statistical analysis included evaluating feature distribution, comparing means (*t*-test) or medians (W-test), and performing a regression analysis and one-way analysis of variance or the Kruskal–Wallis test, as no normal distribution or between-group variance was indicated for the significant differences in the investigated groups. Differences or relationships were considered statistically significant at *p* < 0.05. The research revealed correlations between single crowns, overdenture restoration, bridge restoration, platform switching, prosthetic fracture, CI, and also marginal bone loss where *p* was lower than 0.05. However, the corticalization phenomenon itself has not yet been fully explored. The findings suggest that, depending on the type of prosthetic restoration, the corticalization index may correlate with marginal bone loss or not. Further research is necessary, as the index is suspected to not be homogeneous.

## 1. Introduction

The placement of dental implants and their successful healing after a few months do not mark the end of treatment. A subsequent and equally critical step is the prosthetic restoration of the dental implant. Given the options, treatments can range from single crowns and splinted crowns to dental bridges and overdentures, depending on the patient’s needs. However, this step might not be the last if post-restoration complications arise, potentially leading to periimplantitis and marginal bone loss near the implants, which can ultimately result in implant loss [1,2].

Currently, the condition of periimplantitis, characterized by both radiological and clinical symptoms, is the primary method for detecting marginal bone loss and predicting implant failure. Unfortunately, some symptoms manifest too late, making it impossible to save the implant. Both surgical and non-surgical treatments may prove ineffective if symptoms are identified too late [3]. Radiological examinations, such as single radiographic (RTG) images, offer a minimally invasive method that can provide extensive data for dental surgeons [4]. Sometimes, changes are detectable on RTG images before clinical symptoms become apparent.

Utilizing dental RTG images allows for the calculation of texture features that can describe the image at the pixel level. These texture features can be employed to distinguish between types of bones (cortical or trabecular). The same features were used to develop the Corticalization Index, which quantifies how bone changes or rebuilds throughout the healing period [5,6,7,8]. The corticalization phenomenon is still not well understood and requires further research.

Before implant loss or failure, certain symptoms manifest. What if there were an index that could predict failure before clinical signs appeared? The aim of this study is to present, analyze, and evaluate a new bone index that could potentially forecast dental implant failure following dental implant restoration.

## 2. Materials and Methods

### 2.1. Inclusion Criteria

Participants were included based on the following criteria: presence of osseointegrated implants after 3 months of healing with prosthetic restoration, age of at least 18 years, bleeding on gingival probing below 20%, probing depth of 3 mm or less, good oral hygiene, regular follow-ups, and two-dimensional radiographs taken during routine checks. Laboratory tests included vitamin levels, ion levels, and hormone levels: parathormone (PTH, normal range 10 to 60 pg/mL), thyrotropin (TSH, normal range 0.23–4.0 µU/mL), calcium in serum (Ca^2+^, normal range 9–11 mg/dL), glycated hemoglobin (HbA1c, normal range below 5%), and vitamin 25(OH)D3 (D3, normal range 31–50 ng/mL). Spine densitometry was also considered, where a T-score indicates bone mineral density (BMD). A normal bone score is greater than −1.0; osteopenia is indicated by values between −1.0 and −2.5; and scores below −2.5 indicate osteoporosis (Table 1).

### 2.2. Exclusion Criteria

Exclusion criteria included: loss of implant after the initial 3-month healing period, absence of X-rays, defective X-ray images upon visual assessment, absence of laboratory tests, uncontrolled internal comorbidities (such as diabetes mellitus, thyroid dishormonoses, rheumatoid disease, and other immunodeficiencies), a history of oral radiation therapy, past or current use of cytostatic drugs, soft and bone tissue augmentation, and low quality or lack of follow-up radiographs (Table 2).

### 2.3. Treatment Procedure

All surgeries and prosthetic treatments were performed by a single dentist (M.K.), adhering to all guidelines and protocols. After the initial 3-month healing period, the implant was uncovered under local anesthesia with Articaine + Adrenaline 1:100,000. Standard healing screws were inserted. Impressions were taken two weeks later, following soft tissue healing, and prosthetic restorations were applied, consisting of titanium abutments and crowns made of ceramic and CrCo alloy. The observation period lasted for 5 years.

### 2.4. Data Acquisition

Standardized intraoral radiographs were taken immediately after the surgery (0 M) and after 5 years of observation (5 y). Radiographs were captured using the DIGORA OPTIME radiography system (TYPE DXR-50, SOREDEX, Helsinki, Finland). The RTG images were taken in a standardized manner with the following settings: 7 mA, 70 mV, and 0.1 s. The focus apparatus was provided by Instrumentarium Dental, Tuusula, Finland. Positioners ensured that images were taken repeatedly with a 90° angle of the X-ray beam to the surface of the phosphor plate.

Radiologically recorded peri-implant bone structure was studied through digital texture analysis using the previously proposed version 1 of the Corticalization Index (CI) [9,10]. It consists of the product of a measure that evaluates the number of long series of pixels of similar optical density with the mean optical density of the studied site (in the numerator) and the magnitude of the chaotic arrangement of the texture pattern, i.e., differential entropy (in the denominator).

A total of 2196 samples were divided into groups based on the type of prosthetic restoration: single crowns, splinted crowns, bridges, and overdentures, also considering periodic control. The corticalization index and marginal bone loss were measured and compared near the implant neck on the day of surgery and after five years for single crowns, splinted crowns, bridges, and overdentures. These parameters were also measured and compared in cases of platform switching (PS) and varying bridge lengths. Methods of crown fixation (cementation and screwing) were also considered. The retention loss and multiple instances of retention loss in prosthetic restorations were calculated depending on the corticalization and marginal bone loss near the implant neck. Additionally, fractures in prosthetic restorations were analyzed.

Marginal bone loss (MBL) was measured on radiological images [11] (Figure 1). The texture of X-ray images was analyzed using MaZda 4.6 freeware developed by the University of Technology in Lodz [6,12] to assess measures of corticalization in the peri-implant environment of trabecular bone (representing original bone before implant-related alterations) and soft tissue (indicative of marginal bone loss). MaZda provides both first-order (Mean Optical Density) and second-order (Differential Entropy: DifEntr; Long-Run Emphasis Moment: LngREmph) data. As the second-order data are provided for four directions in the image and the present study did not focus on directional features, the arithmetic mean of these four primary data points was used for further analysis. Regions of interest (ROIs) were marked near the neck area (Figure 2) and normalized (μ ± 3σ) to have the same mean (μ) and standard deviation (σ) of optical density within the ROI. To further reduce noise, data were limited to 6 bits. For analysis in a co-occurrence matrix, a spacing of 5 pixels was chosen. In the formulas that follow, *p*(*i*) is a normalized histogram vector (i.e., histogram entries divided by the total number of pixels in the ROI), *i* = 1, 2, …, and *Ng* denotes the number of optical density levels. The Mean Optical Density feature (a first-order feature) was calculated as follows:$$Mean\;Optical\;Density=\sum_{i=1}^{N_{g}}ip(i)$$

Second order features:$$DifEntr=-\sum_{i=1}^{Ng}p_{x-y}\left(i\right)log(p_{x-y}(i))$$
where Σ is the sum, *Ng* is the number of levels of optical density in the radiograph, *i* and *j* are the optical density of pixels that are 5 pixels distant from one another, *p* is the probability, and log is the common logarithm [13]. The differential entropy calculated in this way is a measure of the overall scatter of bone structure elements in a radiograph. Its high values are typical for cancellous bone. Next, the last primary texture feature was calculated:$$LngREmph=\frac{\sum_{i=1}^{Ng}\sum_{k=1}^{Nr}k^{2}p(i,k)}{\sum_{i=1}^{Ng}\sum_{k=1}^{Nr}p(i,k)}$$
where Σ is the sum, *Nr* is the number of series of pixels with density level *i* and length *k*, *Ng* is the number of levels for image optical density, *Nr* is the number of pixels in the series, and *p* is the probability [14,15]. This texture feature describes thick, uniformly dense, radio-opaque bone structures in intra-oral radiograph images.
$$CI=\frac{LngREmph\cdot Mean\;Optical\;Density}{DifEntr}$$

Statistical analysis included feature distribution evaluation, mean (*t*-test) or median (W-test) comparison, regression analysis, and a one-way analysis of variance or the Kruskal–Wallis test as indicated by non-normal distribution or between-group variance on significant differences in the investigated groups. Differences or relationships were assumed to be statistically significant at *p* < 0.05. Statgraphics Centurion version 18.1.12 (StatPoint Technologies, Warrenton, VA, USA) was used for statistical analyses.

The limitation of the study is that the laboratory tests were not carried out during the observation period.

## 3. Results

### 3.1. Single Crowns

The study revealed that the initial corticalization index was higher in the cases of 1 and 3 neighboring single crown restorations, respectively 149.55 ± 88.62 and 261.41 ± 47.52, where *p* was lower than 0.05. After 5 years of observation, the corticalization index was significantly higher for 1 single crown, at 194.70 ± 189.08, where *p* was lower than 0.05. In the case of 3 single crowns, the CI was 239.73 ± 70.87, and *p* > 0.05. Marginal bone loss correlated with single crowns only initially (mean 0 ± 1.03 mm), where *p* was lower than 0.05, and disappeared after 5 years (mean 0 ± 1.22 mm) of observation, *p* > 0.05. The study revealed a statistically significant difference between 1 and 3 single crowns in the case of corticalization in the initial period, between 0 and 1 single crown in the case of corticalization after 5 years, and between 0 and 1 single crown in the case of marginal bone loss in the initial period (Table 3) (Figure 3).

### 3.2. Splinted Crowns

In the case of splinted crowns, there was no statistical significance initially or after 5 years of observation concerning the corticalization index. However, a relationship was noted between marginal bone loss after 5 years in 2 and 6 splinted crowns, respectively 0.14 ± 1.26 mm and 0.00 ± 0.00 mm (Table 4) (Figure 4).

### 3.3. Overdenture Restoration

The corticalization index was initially higher in the case of overdenture restorations compared to non-overdentures and remained higher 5 years after observation, at 185.46 ± 162.46 and 359.91 ± 248.70, respectively, where *p* was lower than 0.05. After 5 years, marginal bone loss was lower in cases of non-overdenture restorations compared to overdentures, respectively 0 ± 1.24 mm and 0.5 ± 1.47 mm, which was statistically significant (Table 5) (Figure 5).

### 3.4. Bridge Restoration

The corticalization index increased in cases where bridges were used compared to non-bridge restorations. Initially, the CI for bridges and non-bridges was, respectively, 172.03 ± 208.84 and 163.41 ± 112.99. After 5 years of observation, it was, respectively, 250.96 ± 165.89 and 210.88 ± 187.64, with *p* values lower than 0.05 in both cases. Marginal bone loss after 5 years was higher in cases with bridge restorations compared to non-bridges, respectively 0.00 ± 1.3 mm and 0.00 ± 1.24 mm, with a *p* value lower than 0.05 (Table 6) (Figure 6).

### 3.5. Presence of Platform Switching (PS)

This study also revealed that the CI in cases of platform switching presence or absence, after 5 years of observation, was higher and changed from the initial values, respectively, from 155.50 ± 95.73 to 196.50 ± 139.84 (PS) and from 170.65 ± 157.85 to 227.23 ± 190.46 (non-PS), indicating that the increase in the CI was statistically significant when the *p* value was lower than 0.05. Marginal bone loss after the observation period was higher in the cases of non-platform switching implants compared to implants with PS, respectively 0 ± 1.29 mm and 0 ± 1.10 mm, but this was not statistically significant, *p* > 0.05 (Table 7) (Figure 7).

### 3.6. Cemented vs. Screwed Prosthetic Restoration

An increase in the CI was observed in cases of prosthetic restorations that were cemented compared to screwed ones, from 162.74 ± 147.50 at baseline to 218.57 ± 179.20 at 5 years and from 168.99 ± 96.99 at baseline to 194.14 ± 71.81 at 5 years, respectively. After 5 years of observation, there was a weak but statistically significant difference in crown fixation methods, where the *p* value was 0.05; however, no statistical difference was found between them. MBL after 5 years of observation for cemented crowns was 0.00 ± 1.26 mm and for screwed crowns was 0.00 ± 1.29 mm, which was not statistically significant, *p* > 0.05 (Table 8) (Figure 8).

### 3.7. Presence of Retention Loss in Prosthetic Restoration

In this study, the corticalization index (CI) increased in both cases, with and without retention loss of prosthetic restoration, from 163.75 ± 127.30 to 206.92 ± 169.88 where retention loss was present and from 167.51 ± 150.67 to 224.91 ± 186.13 where retention loss was not observed. After 5 years of observation, the study revealed a statistically significant change in the CI between these two groups, where the *p* value was lower than 0.05. Marginal bone loss after 5 years was 0.00 ± 1.5 mm where retention loss occurred at least once, and 0.00 ± 1.16 mm where retention loss was not recorded, with no statistical significance (*p* > 0.05) (Table 9) (Figure 9).

### 3.8. Bridge Length

The study also analyzed the length of bridge restorations and their effects on the corticalization index and marginal bone loss. There was a correlation between the corticalization index value and bridge length. After 5 years of observation, the *p* value was lower than 0.05, with the most significant difference observed between single crowns and 3-point bridges, with CI values of 211.14 ± 189.14 and 282.20 ± 170.30, respectively. Initially, the CI was lower, at 164.23 ± 113.02 for single crowns and 183.00 ± 153.16 for 3-point bridges. The correlation between marginal bone loss (MBL) and bridge length was also analyzed and showed statistically significant changes after 5 years, with the greatest differences observed in groups with 0–3, 3–6, 4–6, 6–7, 6–9, and 6–10 point bridges. MBL values for these groups were: single crowns—0 ± 1.24 mm; 3-point bridges—0 ± 0.91 mm; 4-point bridges—0 ± 1.00 mm; 6-point bridges—1.5 ± 2.54 mm; 7-point bridges—0 ± 0.00 mm; 9-point bridges—0 ± 0.00 mm; and 10-point bridges—0 ± 0.94 mm (Figure 10).

### 3.9. Multiple Retention Losses

Research also investigated the correlation between multiple retention losses of prosthetic restorations and the corticalization phenomenon, as well as marginal bone loss. Statistical analysis revealed no correlation between multiple retention losses and the corticalization index in the initial period. Additionally, there was no correlation between multiple retention losses and the appearance of marginal bone loss in both cases, with no statistical significance (*p* > 0.05) (Figure 11 and Table 8).

### 3.10. Multiple Retention Losses after 5 Years

After 5 years of observation, research showed that a correlation had appeared between multiple retention losses and the corticalization index, with greater differences observed between instances of 2 and 4 times prosthetic retention loss. The CI was 247.73 ± 185.04 and 152.86 ± 93.65, respectively, with a *p* value lower than 0.05, which was statistically significant. There was also a statistically significant difference between multiple retention losses and marginal bone loss after 5 years. The greatest differences were between single retention loss and 2 and 5 times retention loss, with additional differences observed between double retention loss and 3 and 4 times retention loss (Figure 9 and Table 10).

### 3.11. Prosthetic Fracture

The study revealed that the CI in cases where prosthetic fracture was detected was lower compared to no fracture cases, respectively 146.73 ± 47.89 and 167.76 ± 147.47 (*p* < 0.05). After 5 years, in both cases, the CI increased to 209.19 ± 170.13 and 221.68 ± 182.87, respectively. There was no statistical significance, and the *p* value was higher than 0.05. The study also showed that there was no relation between prosthetic fracture and marginal bone loss initially or after 5 years of observation, with a *p* value higher than 0.05 (Table 11) (Figure 12).

## 4. Discussion

The corticalization phenomenon, or corticalization index, is still not well explored in science. The literature reveals only a few studies that describe this phenomenon in dental/implant surgery [8,9,10,16]. Considering the presented research, this phenomenon may reflect various changes in the alveolar bone, both positive and negative, depending on multiple factors.

Studies from 1980 indicated that a marginal bone loss (MBL) of 1–1.5 mm around dental implants in the first year and less than 0.2 mm in subsequent years was acceptable. Is this standard still valid in this century? Recent studies have shown that marginal bone loss depends on factors such as tobacco smoking, the presence or absence of platform switching, cement remnants, overloading in occlusion, poor oral hygiene, and the type of prosthetic restoration [17,18,19,20,21].

The presented research demonstrated that single crown restorations led to lower MBL compared to three single crowns. Additionally, the CI increased in cases of single crowns after 5 years of observation. In cases of splinted crowns, the impact of this type of restoration on bone corticalization was not observed, likely due to the different distribution of masticatory forces transmitted to the bone surrounding the implant [22,23]. If corticalization has a negative effect, then splinted crowns are a beneficial prosthetic solution. This is evidenced by the lack of bone atrophy in the implant neck area despite an increase in the CI. This further indicates that the corticalization phenomenon is not homogeneous. Single crowns remain one of the most popular prosthetic restorations and typically result in less MBL compared to implant-supported prostheses. Splinted crowns also tend to preserve marginal bone around dental implants [24,25].

In the case of implant-based removable prosthetic restorations, such as overdentures, MBL was observed, which was also associated with an increasing corticalization index. Removable implant restorations lead to higher marginal bone loss compared to fixed ones [26], likely due to axial forces during mastication. Therefore, the type of prosthetic restoration, such as overdentures, impacts bone changes such as corticalization and, subsequently, marginal bone loss around the implant neck. At times, the appearance of corticalization may signal impending bone atrophy [27,28].

Bridges with varying lengths were also correlated with changes in the corticalization index and MBL. This restoration type showed an impact on bone rebuilding near the dental implant neck. Bridges with six points exhibited the least favorable mechanical properties compared to shorter or longer ones. Bridge restorations generally revealed better marginal bone preservation compared to overdentures, which is also confirmed in the literature [21].

The presence of platform switching (PS) was also evaluated in this study. The presence or absence of PS did not impact MBL around dental implants. However, an increase in the CI was observed in cases without PS. The occurrence of PS seems to mitigate the adverse effects of increasing corticalization over time. The construction of dental implants may influence bone remodeling near the implants, with studies showing that PS is associated with bone preservation near the implants [27,29,30,31,32]. Given that the CI increased with the presence of PS, it can be inferred that this phenomenon positively impacts bone remodeling.

Cement-retained versus screw-retained prosthetic restorations were also evaluated, comparing MBL and changes in the CI. Some publications indicate that cemented restorations had better outcomes post-observation period [33,34,35] compared to screw-retained ones regarding MBL. No study has focused on the corticalization phenomenon itself. Research shows that the CI increased where a cemented solution was chosen. There was no correlation between MBL in both fixation solutions and corticalization. This suggests that the phenomenon is heterogeneous and may depend on many factors still under investigation.

During the observation period, loosening of retention in prosthetic restorations is possible and is often reported as the most common complication [36,37,38,39]. In both scenarios—whether retention loss was observed or not—the CI increased, but it was higher in cases where no retention loss was indicated. Research also showed that MBL was higher where retention loss occurred. Notably, cases without loosening retention led to a higher CI value and a marginally smaller MBL compared to cases where retention loss occurred. In cases with multiple retention losses, changes in the CI were observed; however, the CI was lower where retention losses occurred more frequently, which correlated with a smaller observed MBL. Does this mean that the CI has a positive relationship with bone remodeling around dental implants and can preserve marginal bone? Generally, if a restoration is losing retention, it suggests negative forces are at play, highlighting the heterogeneity of this phenomenon.

The final complication evaluated in this study was prosthetic/abutment fracture, a rare occurrence in clinical practice [1,40]. No correlation was found between prosthetic fractures and marginal bone loss around dental implants. Research also did not reveal any correlation between fractures and the corticalization phenomenon after 5 years. Nevertheless, higher values of the corticalization index were observed in cases where prosthetic fractures were not present. Did the intact reconstructions transmit greater forces into the bone, thereby remodeling it? What effect does bone remodeling/corticalization around the dental implant neck have on the surrounding bone tissue?

## 5. Conclusions

The corticalization phenomenon could serve as a novel, specific indicator and predictor of bone loss. This research has once again demonstrated that this phenomenon is heterogeneous, manifesting in various forms. Significant correlations were observed between different types of prosthetic restorations and changes in bone structure, including corticalization. Further investigation and detailed descriptions of the corticalization phenomenon are necessary to fully understand its implications.

## Figures and Tables

**Figure 1 diagnostics-14-00867-f001:**
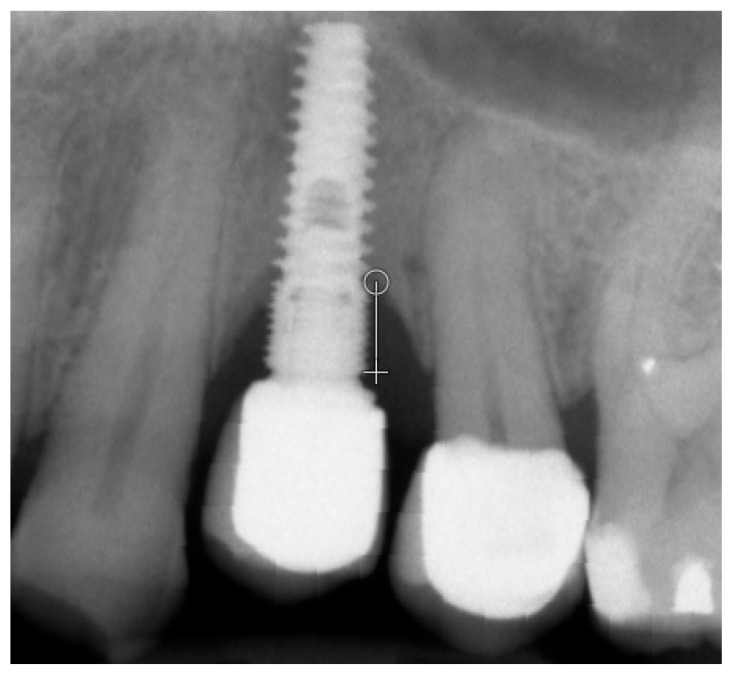
Measurement of marginal bone loss on the radiographic images 5 years after functional loading with a single crown. The white line indicates the implant platform and the circle at the bottom of the bone loss cavity [8].

**Figure 2 diagnostics-14-00867-f002:**
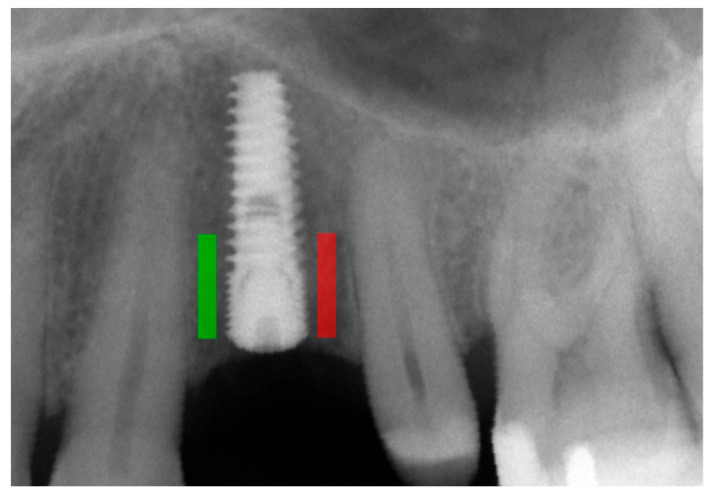
Marking an ROI. ROIs were marked near the implant neck area. Green area—mesial implant neck area; red area—distal implant neck area. Abbreviations: ROI—region of interest.

**Figure 3 diagnostics-14-00867-f003:**
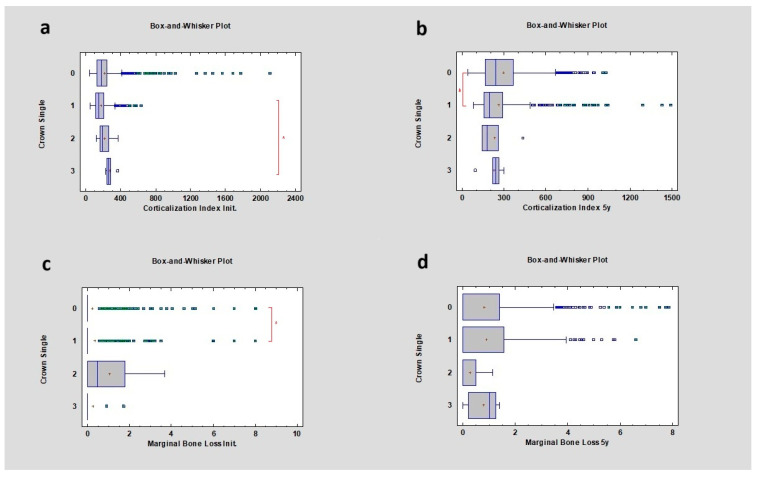
Dependencies for all samples: (**a**) dependence of the number of single crowns on the corticalization index in the initial period; (**b**) dependence of the number of single crowns on the corticalization index after 5 years of observation; (**c**) dependence of the number of single crowns on marginal bone loss in the initial period; (**d**) dependence of the number of single crowns on marginal bone loss after 5 years of observation. The red asterisk describes statistically significant differences.

**Figure 4 diagnostics-14-00867-f004:**
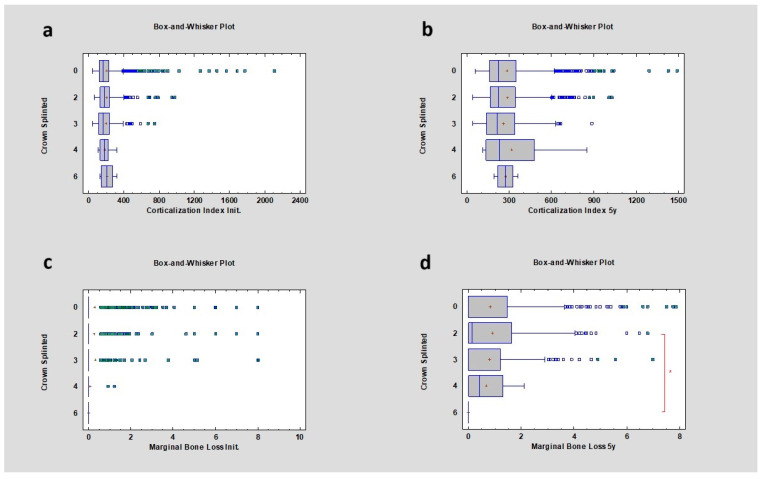
Dependencies for all samples: (**a**) dependence of the number of splinted crowns on the corticalization index in the initial period; (**b**) dependence of the number of splinted crowns on the corticalization index after 5 years of observation; (**c**) dependence of the number of splinted crowns on marginal bone loss in the initial period; (**d**) dependence of the number of splinted crowns on marginal bone loss after 5 years of observation. The red asterisk describes statistically significant differences.

**Figure 5 diagnostics-14-00867-f005:**
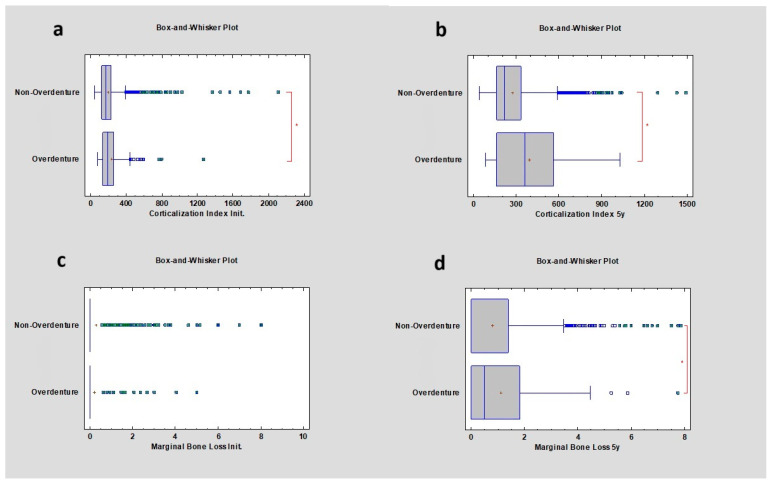
Dependencies for all samples: (**a**) dependence of overdenture or non-overdenture prosthetic restoration on the corticalization index in the initial period; (**b**) dependence of overdenture or non-overdenture prosthetic restoration on the corticalization index after 5 years of observation; (**c**) dependence of overdenture or non-overdenture prosthetic restoration on marginal bone loss in the initial period; (**d**) dependence of overdenture or non-overdenture prosthetic restoration on marginal bone loss after 5 years of observation. The red asterisk describes statistically significant differences.

**Figure 6 diagnostics-14-00867-f006:**
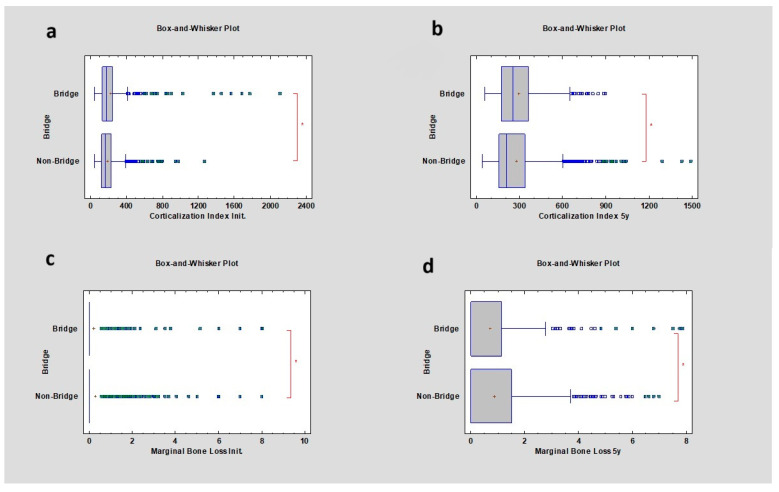
Dependencies for all samples: (**a**) dependence of the presence of a prosthetic bridge or not on the corticalization index in the initial period; (**b**) dependence of the presence of a prosthetic bridge or not on the corticalization index after 5 years of observation; (**c**) dependence of the presence of a prosthetic bridge or not on marginal bone loss in the initial period; (**d**) dependence of the presence of a prosthetic bridge or not on marginal bone loss after 5 years of observation. The red asterisk describes statistically significant differences.

**Figure 7 diagnostics-14-00867-f007:**
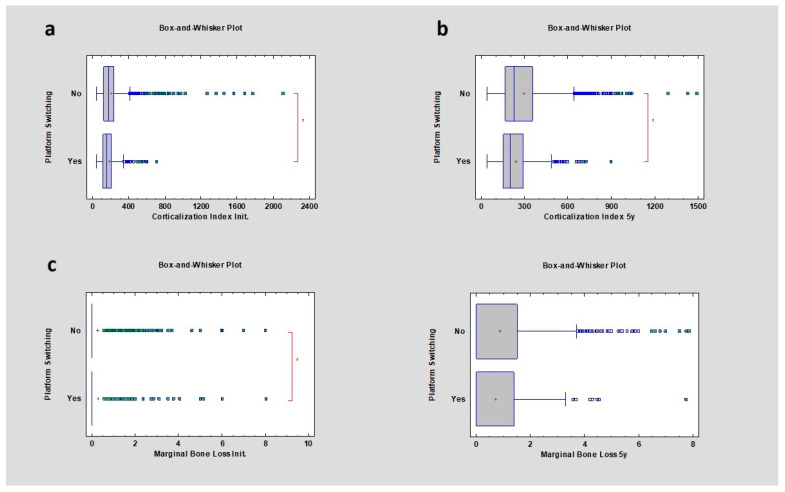
Dependencies for all samples: (**a**) dependence of the presence of platform switching or not on the corticalization index in the initial period; (**b**) dependence of the presence of platform switching or not on the corticalization index after 5 years of observation; (**c**) dependence of the presence of platform switching or not on marginal bone loss in the initial period; (**d**) dependence of the presence of platform switching or not on marginal bone loss after 5 years of observation. The red asterisk describes statistically significant differences.

**Figure 8 diagnostics-14-00867-f008:**
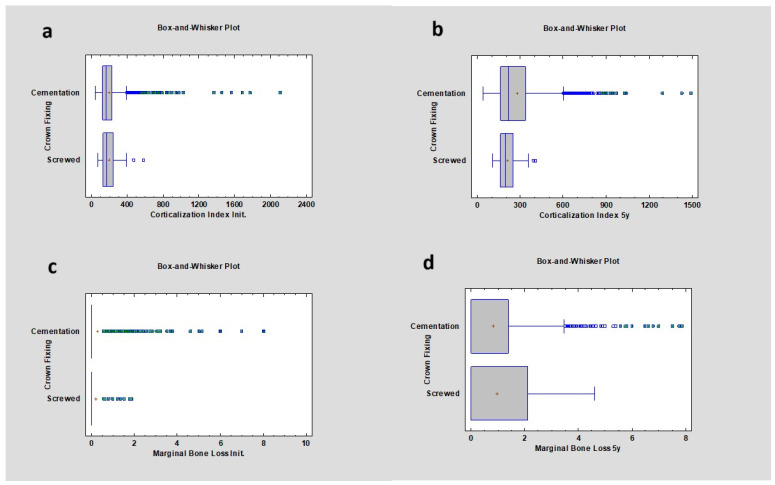
Dependencies for all samples: (**a**) dependence of crown fixing (cemented vs screwed) on the corticalization index in the initial period; (**b**) dependence of crown fixing (cemented vs screwed) on the corticalization index after 5 years of observation; (**c**) dependence of crown fixing (cemented vs screwed) on marginal bone loss in the initial period; (**d**) dependence of crown fixing (cemented vs screwed) on marginal bone loss after 5 years of observation.

**Figure 9 diagnostics-14-00867-f009:**
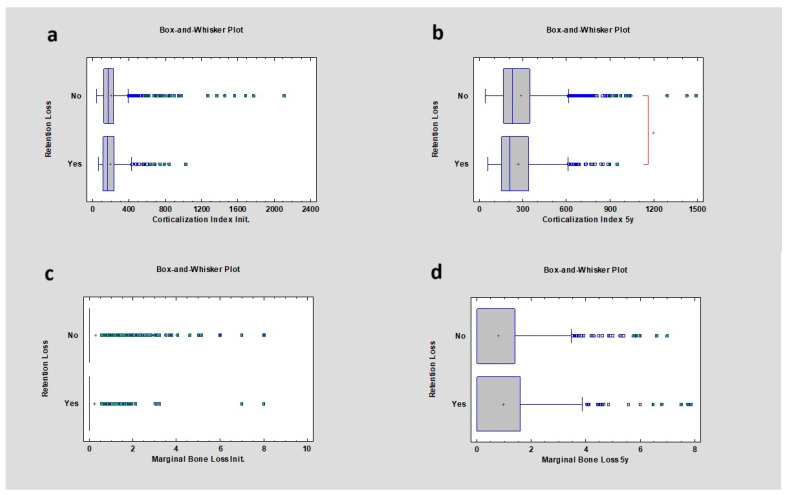
Dependencies for all samples: (**a**) dependence of prosthetic retention loss on the corticalization index in the initial period; (**b**) dependence of prosthetic retention loss on the corticalization index after 5 years of observation; (**c**) dependence of prosthetic retention loss on marginal bone loss in the initial period; (**d**) dependence of prosthetic retention loss on marginal bone loss after 5 years of observation. The red asterisk describes statistically significant differences.

**Figure 10 diagnostics-14-00867-f010:**
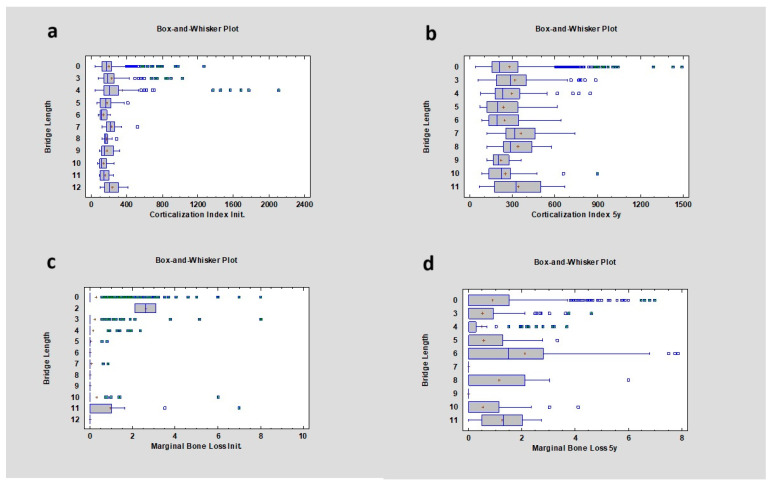
Dependencies for all samples: (**a**) dependence of bridge length on the corticalization index in the initial period; (**b**) dependence of bridge length on the corticalization index after 5 years of observation; (**c**) dependence of bridge length on marginal bone loss in the initial period; (**d**) dependence of bridge length on marginal bone loss after 5 years of observation.

**Figure 11 diagnostics-14-00867-f011:**
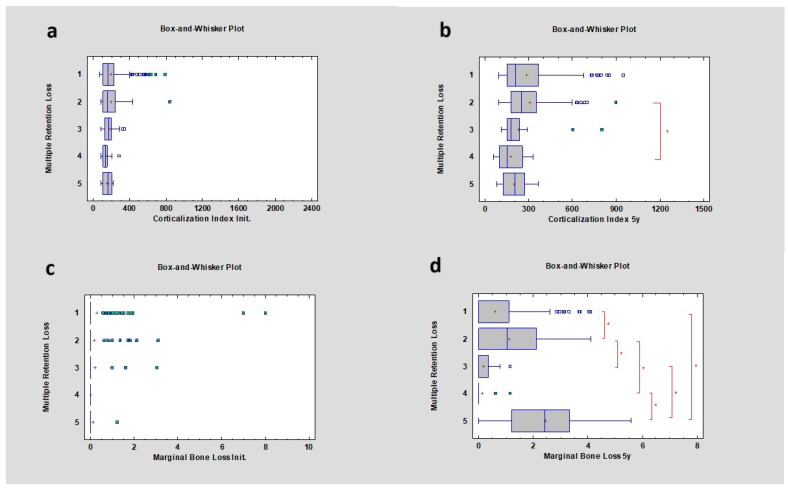
Dependencies for all samples: (**a**) dependence of multiple retention losses on the corticalization index in the initial period; (**b**) dependence of multiple retention losses on the corticalization index after 5 years of observation; (**c**) dependence of multiple retention losses on marginal bone loss in the initial period; (**d**) dependence of multiple retention losses on marginal bone loss after 5 years of observation. The red asterisk describes statistically significant differences.

**Figure 12 diagnostics-14-00867-f012:**
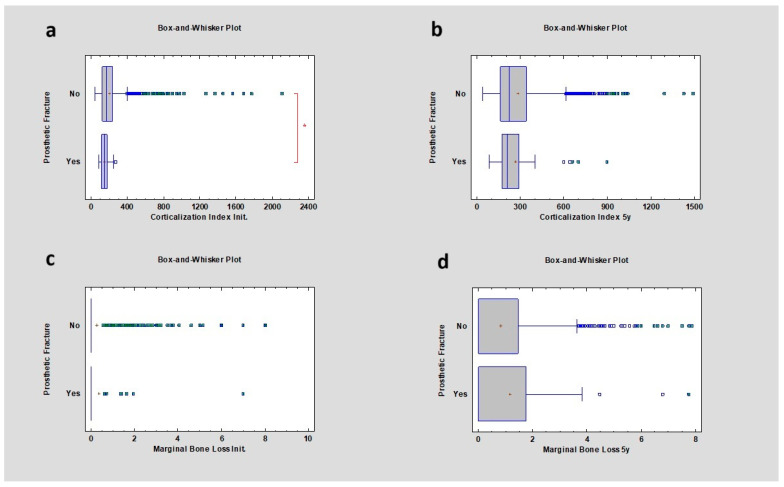
Dependencies for all samples: (**a**) dependence of prosthetic fractures on the corticalization index in the initial period; (**b**) dependence of prosthetic fractures on the corticalization index after 5 years of observation; (**c**) dependence of prosthetic fractures on marginal bone loss in the initial period; (**d**) dependence of prosthetic fractures on marginal bone loss after 5 years of observation. The red asterisk describes statistically significant differences.

**Table 1 diagnostics-14-00867-t001:** Inclusion criteria for the research.

Inclusion Criteria
18 years of age
Bleeding on gingival probing < 20%
Probing depth ≤ 3 mm
Good oral hygiene
Regular follow ups
Two dimensional radiographs taken during the regular check
Laboratory test: PTH, where norm is 10–60 pg/mL; TSH, where norm is 0.23–4.0 µU/mL; Calcium in serum (Ca^2+^), where norm is 9–11 mg/dL; HbA1c, where norm is <5%; Vitamin 25(OH)D3 (D3), where norm is 31–50 ng/mL
Spine densitometry
Smoking 1 or more cigarettes per day

**Table 2 diagnostics-14-00867-t002:** Exclusion criteria for the research.

Exclusion Criteria
Lack of X-rays
Defective X-ray images in the visual assessment
Lack of laboratory tests
Uncontrolled internal co-morbidity: Diabetes mellitus Thyroid dishormonoses Rheumatoid disease Other immunodeficiencies
A history of oral radiation therapy
Past or current use of cytostatic drugs
Soft and bone tissue augmentation
Low quality or lack of follow-up radiographs

**Table 3 diagnostics-14-00867-t003:** Values for marginal bone loss and the corticalization index in the case of single crowns as prosthetic restoration. Values were calculated for all the implantations. 00 M—the observation period immediately after the implantation; 5 y—the observation period 5 years after the implantation.

	Observation Period	Corticalization	*p* Value	Marginal Bone Loss	*p* Value
1 single crown	00 M	149.55 ± 88.62	*p* < 0.05	0 ± 1.03 mm	*p* < 0.05
	5 y	194.70 ± 189.08	*p* < 0.05	0 ± 1.22 mm	*p* = 0.20
3 single crowns	00 M	261.41 ± 47.52	*p* < 0.05	0 ± 0.58 mm	*p* = 0.39
	5 y	239.73 ± 70.87	*p* = 0.20	1 ± 0.57 mm	*p* = 0.32

**Table 4 diagnostics-14-00867-t004:** Values for marginal bone loss in the case of splinted crowns as prosthetic restoration. Values were calculated for all the implantations. 00 M—the observation period immediately after the implantation; 5 y—the observation period 5 years after the implantation.

	Observation Period	Marginal Bone Loss	*p* Value
2 splinted crowns	00 M	0 ± 0.97 mm	*p* = 0.14
	5 y	0 ± 1.31 mm	*p* < 0.05
6 splinted crowns	00 M	0.00 ± 0.00 mm	*p* = 0.14
	5 y	0.00 ± 0.00 mm	*p* < 0.05

**Table 5 diagnostics-14-00867-t005:** Values for marginal bone loss and the corticalization index in the case of overdenture and non-overdenture restorations. Values were calculated for all the implantations. 00 M—the observation period immediately after the implantation; 5 y—the observation period 5 years after the implantation.

	Observation Period	Corticalization	*p* Value	Marginal Bone Loss	*p* Value
Overdenture	00 M	185.46 ± 162.46	*p* < 0.05	0 ± 0.7 mm	*p* = 0.44
	5 y	359.91 ± 248.70	*p* < 0.05	0.49 ± 1.47 mm	*p* < 0.05
Non-overdenture	00 M	164.05 ± 144.87	*p* < 0.05	0 ± 0.96 mm	*p* = 0.44
	5 y	218.57 ± 175.17	*p* < 0.05	0 ± 1.23 mm	*p* < 0.05

**Table 6 diagnostics-14-00867-t006:** Values for marginal bone loss and the corticalization index in the case of bridge restorations compared to non-bridge restorations. Values were calculated for all the implantations. 00 M—the observation period immediately after the implantation; 5 y—the observation period 5 years after the implantation.

	Observation Period	Corticalization	*p* Value	Marginal Bone Loss	*p* Value
Non-bridge	00 M	163.41 ± 112.99	*p* < 0.05	0 ± 0.95 mm	*p* < 0.05
	5 y	210.88 ± 187.64	*p* < 0.05	0 ± 1.24 mm	*p* < 0.05
Bridge	00 M	172.03 ± 208.84	*p* < 0.05	0 ± 0.91 mm	*p* < 0.05
	5 y	250.96 ± 165.89	*p* < 0.05	0 ± 1.30 mm	*p* < 0.05

**Table 7 diagnostics-14-00867-t007:** Values for marginal bone loss and the corticalization index in the case of platform switching presence or absence. Values were calculated for all the implantations. 00 M—the observation period immediately after the implantation; 5 y—the observation period 5 years after the implantation; PS—platform switching; no PS—no platform switching.

	Observation Period	Corticalization	*p* Value	Marginal Bone Loss	*p* Value
No PS	00 M	170.65 ± 157.85	*p* < 0.05	0 ± 0.96 mm	*p* = 0.26
	5 y	227.23 ± 190.46	*p* < 0.05	0 ± 1.29 mm	*p* = 0.15
PS	00 M	155.50 ± 95.73	*p* < 0.05	0 ± 0.86 mm	*p* = 0.26
	5 y	196.50 ± 139.84	*p* < 0.05	0 ± 1.10 mm	*p* = 0.15

**Table 8 diagnostics-14-00867-t008:** Values for marginal bone loss and the corticalization index in the case of the crown fixation method. Values were calculated for all the implantations. 00 M—the observation period immediately after the implantation; 5 y—the observation period 5 years after the implantation.

	Observation Period	Corticalization	*p* Value	Marginal Bone Loss	*p* Value
Cemented	00 M	162.74 ± 147.50	*p* = 0.26	0 ± 0.98 mm	*p* = 0.63
	5 y	218.57 ± 179.20	*p* = 0.05	0 ± 1.26 mm	*p* = 0.52
Screwed	00 M	168.99 ± 96.99	*p* = 0.06	0 ± 0.46 mm	*p* = 0.63
	5 y	194.14 ± 71.81	*p* = 0.05	0 ± 1.29 mm	*p* = 0.52

**Table 9 diagnostics-14-00867-t009:** Values for marginal bone loss and the corticalization index in the case of the presence or absence of prosthetic restoration retention loss. Values were calculated for all the implantations. 00 M—the observation period immediately after the implantation; 5 y—the observation period 5 years after the implantation.

	Observation Period	Corticalization	*p* Value	Marginal Bone Loss	*p* Value
No retention loss	00 M	167.51 ± 150.67	*p* = 0.44	0 ± 0.96 mm	*p* = 0.63
	5 y	224.91 ± 186.13	*p* < 0.05	0 ± 1.16 mm	*p* = 0.52
Retention loss	00 M	163.75 ± 127.30	*p* = 0.44	0 ± 0.86 mm	*p* = 0.63
	5 y	206.92 ± 169.88	*p* < 0.05	0 ± 1.5 mm	*p* = 0.52

**Table 10 diagnostics-14-00867-t010:** Values for marginal bone loss and the corticalization index in the case of the presence of multiple retention and multiple retention losses after 5 years. Values were calculated for all the implantations. 00 M—the observation period immediately after the implantation; 5 y—the observation period 5 years after the implantation.

Retention Loss Times	Observation Period	Corticalization	*p* Value	Marginal Bone Loss	*p* Value
1	00 M	160.64 ± 123.28	*p* = 0.56	0 ± 1.06	*p* = 0.53
	5 y	205.265 ± 191.77	*p* < 0.05	0 ± 1.00	*p* < 0.05
2	00 M	153.92 ± 124.62	*p* = 0.56	0 ± 0.52	*p* = 0.53
	5 y	247.73 ± 185.04	*p* < 0.05	1.03 ± 1.15	*p* < 0.05
3	00 M	170.04 ± 74.92	*p* = 0.56	0 ± 0.65	*p* = 0.53
	5 y	174.35 ± 162.25	*p* < 0.05	0 ± 0.32	*p* < 0.05
4	00 M	130.33 ± 47.44	*p* = 0.56	0 ± 0	*p* = 0.53
	5 y	152.86 ± 93.645	*p* < 0.05	0 ± 0.33	*p* < 0.05
5	00 M	157.35 ± 51.78	*p* = 0.56	0 ± 0.35	*p* = 0.53
	5 y	199.91 ± 88.29	*p* < 0.05	2.42 ± 1.72	*p* < 0.05

**Table 11 diagnostics-14-00867-t011:** Values for marginal bone loss and the corticalization index in the case of the presence or absence of prosthetic fracture. Values were calculated for all the implantations. 00 M—the observation period immediately after the implantation; 5 y—the observation period 5 years after the implantation; No PF—no prosthetic fracture; PF—prosthetic fracture.

	Observation Period	Corticalization	*p* Value	Marginal Bone Loss	*p* Value
No PF	00 M	167.76 ± 147.47	*p* < 0.05	0 ± 0.94 mm	*p* = 0.51
	5 y	221.68 ± 182.87	*p* = 0.73	0 ± 1.24 mm	*p* = 0.38
PF	00 M	146.73 ± 47.89	*p* < 0.05	0 ± 1.16 mm	*p* = 0.51
	5 y	209.19 ± 170.13	*p* = 0.73	0 ± 1.8 mm	*p* = 0.38

## Data Availability

The data on which this study is based will be made available upon request at https://www.researchgate.net/profile/Tomasz-Wach, access date: 14 April 2024.

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
