# Peer review of "New Radiological Corticalization Index as an Indicator of Implant Success Rate Depending on Prosthetic Restoration—5 Years of Follow-Up"

_diagnostics, 2024, doi:10.3390/diagnostics14090867_

Round 1

Reviewer 1 Report

Comments and Suggestions for Authors

Dear Authors

The manuscript "Radiological Evaluation of Bone Remodelling of the Dental Implant Neck Area Depending on the Type of Prosthetic Restoration. New Radiological Corticalization Index as an Indicator of Implant Success Rate after Years of Follow-Up" by Tomasz Wach, Jakub Okulski, Rafał Zieliński, Grzegorz Trybek, Adam Michcik and Marcin Kozakiewicz, is an article whose research aimed to evaluate the correlation between IC and bone remodeling using only RTG images. Overall, the manuscript left me with a good impression.

The structure of the manuscript is well designed. The adequate investigation methods are used. There are not missing tables, figures and references.

However, there are several remarks, which should be edited.

In the Abstract section, please follow the MDPI authors' guidelines concerning the abstract structure. Ensure it is presented without headings.

- The English of the whole manuscript should be checked and edited.

Please revise the keywords section, ensuring compatibility with MeSH indexing.

- Sentence punctuation in the introduction is not formatted, (marked in yellow).

- In Materials and methods, a schematic or image of the sample must be added.

- Line 110 (Figure1) I can't find the image

- Line 119 (Figure2) I can't find the image

Line 158 (Figure1) It is referenced again Figure1

- Line 174 (Figure2) It is referenced again Figure2

– In lines 185 and 202 there are some sentences (marked in yellow) that are not clear enough. They should be corrected.

– The conclusions are not clear enough. The conclusions do not correspond to the results obtained.

- The References are not correct. - Please follow the styles recommended for MDPI journals.

Comments on the Quality of English Language

The English of the whole manuscript should be checked and edited

Author Response

Dear Reviewer, thank You for a such positive and detailed revision of our work. We did all suggested corrections and we hope that now whole manuscript will meet Your and Journal requirements for publishing. Thanks to Your suggestions, our research sound and look more scientific. Below are Your comments and solutions that were applied. Thank You again.

In the Abstract section, please follow the MDPI authors' guidelines concerning the abstract structure. Ensure it is presented without headings - corrected.

- The English of the whole manuscript should be checked and edited – English of the whole manuscript was checked and edited.

- Please revise the keywords section, ensuring compatibility with MeSH indexing- corrected.

- Sentence punctuation in the introduction is not formatted, (marked in yellow).- corrected

- In Materials and methods, a schematic or image of the sample must be added. - corrected

- Line 110 (Figure1) I can't find the image - corrected

- Line 119 (Figure2) I can't find the image - corrected

- Line 158 (Figure1) It is referenced again Figure1 - corrected

- Line 174 (Figure2) It is referenced again Figure2 - corrected

– In lines 185 and 202 there are some sentences (marked in yellow) that are not clear enough. They should be corrected. - corrected

– The conclusions are not clear enough. The conclusions do not correspond to the results obtained. - corrected

  • The References are not correct. - Please follow the styles recommended for MDPI journals. – style of the references usually is corrected by Journal. This was also made in this case but I needed to use old version of the manuscript. Hope Journal will fix the style of references.

Yours Sincerely
Wach Tomasz

Reviewer 2 Report

Comments and Suggestions for Authors

Dear Authors,

Thank you for submitting the manuscript "Radiological evaluation of bone remodeling of the dental implant neck area depending on the type of prosthetic restoration. New radiological corticalization index as an indicator of implant success rate after 5 years follow-up. I carefully read it and here is my feedback:

-The title should not be a paragraph, you need to significantly reduce the number of words.

-Remove the self-citation references or manuscript will be rejected due to unethical behavior (Wach has 7 and Kozakiewicz has 11, together make almost 50% of all your references).

-A major English grammar revision is needed, because some sentences are not well connected.

 -Remove "locked crowns" in line 35, it does not make sense for any meaning and replace "bridges" in line 35-36 for fixed dental prostheses. Please review your professional dental grammar. You can use the Glossary of Prosthodontic Terms to use the proper terminology.

-Second paragraph, please do not start a paragraph with a question, review your English grammar.

-Lines 51-53 have two sentences to make a full paragraph, please do not make short paragraphs.

-Make a small table describing the inclusion and exclusion criteria for the study.

-Clarify, you requested permission to perform the radiographic study in 2011 and you are trying to publish in 2024, why did it take so long ?  implant therapy has significantly changed during that time period.

-What type of radiographs you evaluated (periapical or bite wings or both) ?

-Please include radiographs showing the findings.

-"Surgery Procedure" but you start describing the prosthetic treatment, please re-write the article to make sense.

-Describe what type of material (abutment and crown, two connections or single) was use for the prosthetic part.

-Discussion is extremely weak and short. You are evaluating several types of implant prostheses but describing the differences according to the materials and surface (polishing) treatment provided.

-You never mention anything regarding the occlusal scheme initially, during and after the evaluation of the study.

-This study needs to mention the several limitations that it has.

Very poor structure for the manuscript.

Limited data describing the materials and methods.

English grammar is extremely poor that some times it is impossible to understand the connection between sentences.

-Make the conclusion shorter (2-3 sentences no more).

Comments on the Quality of English Language

The entire English grammar needs to be revised, because it is hard to understand what the authors mean.

Author Response

Dear Reviewer,

thank You for Your opinion and detailed revision of our work. We hope that all suggested corrections were applied and now whole manuscript will meet Your and Journal requirements for publishing. Thanks to Your suggestions, our research sound and look more scientific. All Your suggestions were applied and corrections were done. Looking for Your positive opinion about the research .

Yours Sincerely
Wach Tomasz

Reviewer 3 Report

Comments and Suggestions for Authors

 I appreciated the study design and the utility of corticalization index

Marginal bone loss (MBL ) was also  measured on radiological images.

I appreciated the statistical analyze  and the clinical concept about the correlation between implants and  the type of prosthetic  restoration 

Author Response

Dear Reviewer, thank You for a such positive revision of our work. We did all suggested corrections by other Reviewers and now we hope that whole manuscript will meet Journal requirements for publishing. Thank You again.

Round 2

Reviewer 1 Report

Comments and Suggestions for Authors

Dear authors

The manuscript was sufficiently improved to warrant publication in Diagnostics and all changes were made as suggested.

Reviewer 2 Report

Comments and Suggestions for Authors

The revision has significantly improved the manuscript. The authors have addressed all the concerns. Therefore, I recommend the manuscript for publication.